# Armband Sensors Location Assessment for Left Arm-ECG Bipolar Leads Waveform Components Discovery Tendencies around the MUAC Line

**DOI:** 10.3390/s22197240

**Published:** 2022-09-24

**Authors:** Omar Escalona, Sephorah Mukhtar, David McEneaney, Dewar Finlay

**Affiliations:** 1School of Engineering, Ulster University, Newtownabbey BT37 0QB, UK; 2Craigavon Area Hospital, Portadown BT63 5QQ, UK

**Keywords:** bipolar cardiac electrograms, arm-ECG monitoring, wearable sensor systems, signal averaged ECG, QRS detection performance, bipolar arm-ECG P-wave vectors, MUAC, BMI

## Abstract

Sudden cardiac death (SCD) risk can be reduced by early detection of short-lived and transient cardiac arrhythmias using long-term electrocardiographic (ECG) monitoring. Early detection of ventricular arrhythmias can reduce the risk of SCD by allowing appropriate interventions. Long-term continuous ECG monitoring, using a non-invasive armband-based wearable device is an appealing solution for detecting early heart rhythm abnormalities. However, there is a paucity of understanding on the number and best bipolar ECG electrode pairs axial orientation around the left mid-upper arm circumference (MUAC) for such devices. This study addresses the question on the best axial orientation of ECG bipolar electrode pairs around the left MUAC in non-invasive armband-based wearable devices, for the early detection of heart rhythm abnormalities. A total of 18 subjects with almost same BMI values in the WASTCArD arm-ECG database were selected to assess arm-ECG bipolar leads quality using proposed metrics of relative (normalized) signal strength measurement, arm-ECG detection performance of the main ECG waveform event component (QRS) and heart-rate variability (HRV) in six derived bipolar arm ECG-lead sensor pairs around the armband circumference, having regularly spaced axis angles (at 30° steps) orientation. The analysis revealed that the angular range from −30° to +30°of arm-lead sensors pair axis orientation around the arm, including the 0° axis (which is co-planar to chest plane), provided the best orientation on the arm for reasonably good QRS detection; presenting the highest sensitivity (Se) median value of 93.3%, precision PPV median value at 99.6%; HRV RMS correlation (p) of 0.97 and coefficient of determination (R^2^) of 0.95 with HRV gold standard values measured in the standard Lead-I ECG.

## 1. Introduction

Cardiovascular diseases (CVDs) are most common cause of death worldwide [1]. It is estimated that CVDs account for 31.5% of global deaths. Sudden Cardiac death accounts for 40% of all CVD deaths [2]. The most common cardiac arrhythmia is atrial fibrillation. Almost 800,000 people in the UK have atrial fibrillation (AF) [3] which is associated with an increased risk of stroke and heart failure, particularly in the elderly population. Most patients with atrial fibrillation (AF) are unaware of its presence, i.e., asymptomatic. Additionally, as revealed by recent studies, defects in the autonomic nervous system (ANS), which involves the sympathetic, parasympathetic, and intrinsic neural networks, have been associated with the pathogenesis of AF [4], such abnormalities can be explored using heart rate variability (HRV) analysis on relatively long-term ECG recordings, by measuring the time intervals between heartbeats over a period of time, and it represents the equilibrium between the parasympathetic nervous system (PNS) and sympathetic nervous system (SNS) [5,6]. Early detection of AF onset could enable appropriate interventions, e.g., anticoagulation, aimed to reduce the risk of stroke. However, early detection of such arrhythmias can be problematic as they tend to be episodic and short-lived, lasting only a few seconds. Such episodes are unlikely to be seen on regular short ‘snapshot’ type ECG recordings and require long-term cardiac monitoring to be detected; this presents a considerable challenge [7]. Current technologies for recording cardiac arrhythmias include implantable loop recorders (ILR) for long-term and Holter monitors for recording over shorter intervals [3]. These different approaches have advantages and disadvantages. Holter devices use adhesive patches placed on the chest and arms, which can be restrictive to the patient in daily life, while the adhesive patches can cause skin irritation when worn for prolonged periods. Implanted loop recorders are expensive and require a surgical procedure necessitating hospital admission for surgical implantation [7].

Current advances in wireless technology have impacted health care significantly, and electrocardiogram (ECG) monitoring devices are no different. Improvements in traditional ECG recording and wireless technology have resulted in portable, interconnected and smart ECG recorders and eHealth networking [8,9,10,11]. The development of cost-effective non-invasive long-term ECG monitoring alternatives will reduce mortality in subjects at high risk for SCD [12]. New bipolar or differential leads definitions and advantageous position of ECG body electrodes offering improved ECG arrhythmias diagnosis (e.g., AF), or enhanced ECG events waveform feature extraction have been investigated [13]. Recent approaches to wearable ECG devices include T-shirts, vests, belts, patches and armbands [7]. Through previous research [7,14,15,16,17,18], it has been confirmed that bipolar or differential ECG signals recorded on the upper arm using an armband sensor system are convenient for long-term ECG monitoring, as with this, no electrodes adhesive, irritating gels and annoying cables are eliminated; making them comfortable to wear. Advanced denoising and signal processing techniques permit extraction of clinically useful information from a far-field, low amplitude arm ECG signals despite considerable noise from muscle artefact [12,17].

In the foregoing work by this research team, it has been devised a novel unobtrusive upper-arm wearable armband prototype ECG monitoring system (WAMECG) for the left upper-arm [14,16] or Mid-Upper Arm Circumference (MUAC) anatomical position. The prototype device uses the valuable insight from our previous studies for extracting meaningful ECG information using far-field bipolar arm-ECG leads for cardiac rhythm monitoring. The angular disposition around the upper-arm of the bipolar ECG electrodes pair (diametrically opposite) embedded in the armband device is an important question, as the research team has reported that certain radial axis orientation present better signal quality than other axes [18]. The far-field ECG data recorded on the arm is prone to high noise due to muscle artefact, and identifying proper rotational positioning of the armband device can significantly increase the chances of recording better quality ECG waveforms and their reliable detection from the distal arm area. Armband ECG recording devices use bipolar electrodes pair sensing approach, thus, a priori best performing axial rotation position of the electrodes pair around the MUAC line will help address the question of determining the electrodes pair axial orientations; offering higher ECG signal quality and strength tendency. Such knowledge would effectively aid enhancing the ECG monitoring quality performance of the armband device.

This retrospective study uses a subset of the WASTCArD arm-ECG mapping database [12,17] to identify derived bipolar ECG sensor pairs (diametrically opposite) axial orientation positions around the left MUAC line, confined in an armband, and stablish some predominant tendency for better electrocardiographic signal recording. Currently, there is a paucity of knowledge regarding this in the current literature; most of which has been reported by our research team since 2013 [7,12,14,17]. Thus, knowledge on a preferred range of arm-ECG sensor pairs axial angular orientation with respect to anatomical reference points, including the chest plane and arm axilla, is investigated and provided. 

In this article, Section 2.1 describes the dataset selection, then Section 2.2 and Section 2.3 present the definition of a set of six bipolar arm lead axes derived from unipolar arm-ECG recordings, at estimated regular angle intervals around the MUAC line, and the methods used to assess their bipolar arm-ECG relative quality and predominant type of ECG waveform component being discovered. Section 2.4 and Section 2.5 present the analysis and signal processing methods used in this investigation to yield a rough angular map around the upper-arm, along the MUAC line, identifying the most favourable axial angular orientation for ECG sensing electrodes pair in the armband. For this, the relative ECG signal quality of the arm leads were assessed by useful metrics for relative signal ratios including: signal-to-noise ratio (SNR), P-wave to QRS (%) signal level ratio (PqrsR) and T-wave to QRS (%) signal level ratio (TqrsR), after some arm ECG signal enhancement process such as signal averaging [14,17]. Also presented there, are assessment methods for ECG event (e.g., QRS complex) detection performance of each derived arm-lead axis, by means of sensitivity (Se) and positive predictive value (PPV or detection precision) metrics, using a standard QRS detection algorithm, such as Pan-Tompkins [19]. Then, Section 2.4.5 presents methods for an additional long-term monitoring clinical application of HRV metrics performance analysis on the six derived bipolar upper-arm leads, using a widely adopted time-domain HRV metric, such as the root mean square of normal heartbeat intervals (RMS) [5], and are correlated with HRV values measured from the standard chest limb Lead-I (as the gold standard) in each subject case. Analysis results figures and their discussion are presented in Section 3 and Section 4. There, the main contributions in this article are presented; the arm leads with highest signal ratio values, best QRS detection sensitivity and precision, highest HRV metric Pearson’s correlation coefficient (p) and highest trendline coefficient of determination (R^2^), would form a clique of arm leads circular range of electrode pairs radial axis orientations around the arm-ECG recording device having predominant tendency for better quality Arm-ECG signal recording. 

This investigation provides a first angular map around the MUAC line presenting axial angular orientation of ECG sensing electrodes pair with best prospect for arm-ECG signal recording. In particular, this study reveals that for reasonably good QRS detection performance, the clique of arm leads corresponds to the angular range from −30° to +30°of arm-lead sensors pair axis orientation around the arm, including the 0° axis, which is co-planar to chest plane.

## 2. Materials and Methods

### 2.1. Extracted Data Set

The clinical arm-ECG dataset used to retrospectively investigate the best angular position of bipolar electrode pairs axes around the left MUAC line was extracted from the WASTCArD arm-ECG mapping clinical database recorded for previous studies [14,17,18]. This arm-ECG database was recorded at Craigavon Area Hospital in Portadown (Northern Ireland); under approval of the local medical research ethics filter committee for Northern Ireland: ORECNI (Office for Research Ethics Committees Northern Ireland) reference 16/NI/0158, and IRAS (Integrated Research Application System, registered project ID:203125, dated: 21 September 2016 [17]. For each case, the ECG data was recorded for up to 8 min in a controlled environment, when the subjects were at rest lying or sitting in a relaxed position. Patients’ demographics included their height and weight, which are used to calculating the associated body mass index (BMI, in Kg/m^2^) and then estimate their associated mid-upper arm circumference (MUAC) length value (cm). The extracted data includes the derived ECG Lead I, proximally recorded on the chest, using the GND electrode in the upper-arm as reference (not conventionally using a separate RL electrode) and three upper left-arm unipolar ECG channels (Ch8, Ch9, Ch10) in a circumferential line around the arm (MUAC line), recorded with respect the Common reference electrode (“ground”, GND) within the MUAC line, as shown in Figure 1. Thus, the derived proximal chest Lead I is: -(RA); i.e., the unipolar RA channel with inverted polarity, will be referred in this study as the Chest Lead I, or simply Chest Lead, and will be used as the gold standard reference ECG signal; for estimating the activation time feature corresponding to the heart’s true electrophysiological ECG waveform events (QRS complex, P-wave, T-wave, etc.), which in principle, are in synchronism with respective far-field bipolar arm-ECG waveform events occurrence; as they are originated from same dipole source (the heart). Thus, the Chest Lead I reference ECG enables the accurate assessment of different bipolar arm-lead ECG events signal discovery quality and their detection performance assessment when using conventional ECG signal processing techniques [19].

**Figure 1 sensors-22-07240-f001:**
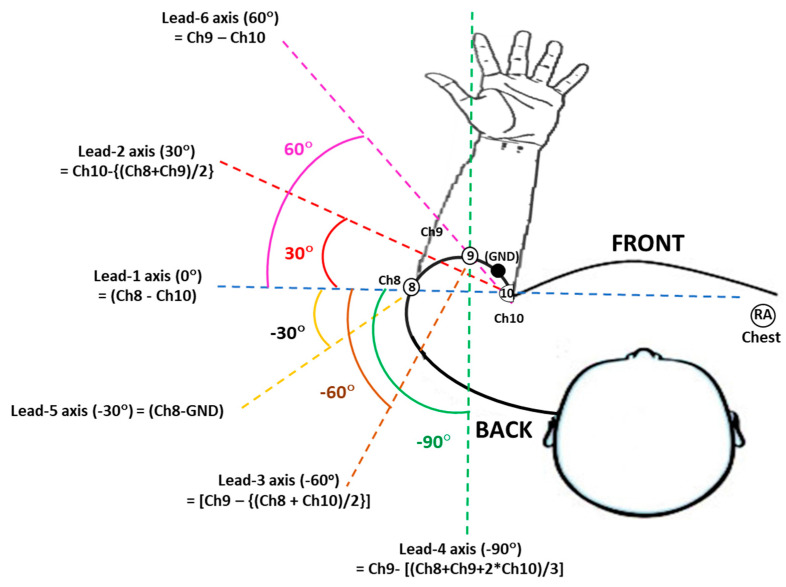
Placement of mid upper-arm channel electrodes on the left arm during the WASTCArD arm-ECG mapping database recordings and derived radial arm axis rotation angles (30° stepwise) of six bipolar-leads around the mid upper-arm circumference (MUAC), as defined in Table 1.

A specific dataset suitable for this retrospective pilot study was extracted from the WASTCArD database based on subjects’ BMI of almost same value; hence, having about same arm diameter, or arm circumference length (MUAC) value. BMI was calculated by dividing the weight of the patient in kg by the square of the height, (BMI = weight (kg)/[Height (m)]^2^) [20,21]. The BMI value of patients correlates to the MUAC value and can be calculated by means of a simple equation presented by Brito et al. in their study on the relation between BMI and MUAC length value [20]. Their study suggests that BMI and MUAC are correlated and MUAC can be used to estimate the BMI value of the subject (and vice versa) using their empirical linear regression equation (Equation (1)), with a related Pearson correlation coefficient of *p* = 0.78. Thus, from the uncertainty associated with such *p* value of the equation, it is estimated that calculations of MUAC values from BMI values using Equation (1) would implicitly be affected by an approximate 5% standard error.
(1) MUAC(cm)=1.1455 (BMI+0.0481)

This empirical line equation relationship between BMI and MUAC indicates that by restricting the BMI values in the included cases sample will also keep their arm size circumference (and diameter) variations within certain tolerable range (preferably about ±2%), yielding to consistent positioning of arm-ECG sensors strap band around the arm from the axilla point (4-electrode strap of fixed length: 18 cm, see next Section), hence, keeping consistency of derived bipolar arm-ECG leads radial axis angular orientation, enabling a fair arm ECG signal quality performance analysis.

The patients’ demographic characteristics of the included cases for this pilot study are summarised in Table 2. The above described subjects sample inclusion criterium led to a subset of 18 subject cases presenting similar BMI around a mean value of 24.9 Kg/m^2^ with a 0.43 SD (see Table 2), which is within a ±2% variation margin of the 18 subjects sample BMI values, and hence, presenting similar upper-arm-circumference (MUAC) values: mean of 28.6 cm with a 0.50 SD (in Table 2). 

### 2.2. Data Acquisition and Sensor System

As described in previous reports [15,16,17,18], the left-arm ECG data was recorded using the BIS-Quatro^TM^ strip of 4 fixed position, Ag-AgCl gelled electrodes, along the linear strip sensor system of fixed open length of 18 cm, having fixed ECG electrodes position along the strip: Ch10 (systematically placed at the axilla point), GND, Ch9 and Ch8 (see Figure 1), and which can be easily wrapped around the upper-arm (anticlockwise, top viewed as in Figure 1), providing systematic case-by-case arm-electrodes positioning, good adhesion and practical connectivity avoiding human errors (crossover misconnection) in channelling data acquisition of the 3 channels and GND, than individually placed ECG electrodes around the arm. This arm-ECG recording method facilitated the larger arm-ECG mapping data collection for the WASTCArD arm-ECG database. 

In the upper arm-ECG 3-channel recording method using the BIS-Quatro^TM^ 4-electrode strip, the proximal electrode corresponding to channel 10 (Ch10), is placed on the inner side of the left arm (near the axilla). The following proximal electrode is the ground electrode (GND). The middle distal electrode (Ch9) sits at a position towards the outer and front side of the upper arm (see Figure 1); the distance between Ch10 and Ch9 on the strap is 6 cm, whereas the distal electrode (Ch8), which can be seen at the farthest end, wraps around the arm and gets placed towards the mid-back side of the upper arm. The interelectrode distance between Ch10 and Ch8 being of 14 cm. Thus, the arc length of Ch9 and Ch8 with respect to Ch10 position (0° reference angle, referred to the chest plane and arm axilla) was used to determine the angles formed by electrodes Ch9 and Ch8 anticlockwise around the arm. The formulae Equations (2) and (3) were used. The Ch8 − Ch9 − Ch10 electrode points form a scalene triangle around the arm.
Angle°Ch8 = (360° (Arc8))/MUAC = (360° (14))/MUAC(2)
Angle°Ch9 = (360° (Arc9))/MUAC = (360° (6))/MUAC (3)

#### 2.2.1. Front-End Data Acquisition System and Signal Characteristics

The BioSemi Active II (BioSemi, Amsterdam) multichannel data acquisition system for clinical research (CE marked) was used in the WASTCArD digital recording protocol. It was configured for a wide bandwidth biopotential signals data capture: 0 Hz–1024 Hz, with the sampling frequency set at 2048 Hz. The data was saved using a BDF file format and was then converted to a MS Excel format ready for processing or exporting. The ECG data files recorded by the BioSemi Active II system were exported and processed in MathWorks MATLAB ver. R2020b, Massachusetts, USA [22].

#### 2.2.2. Pre-Filtering

Due to the wide bandwidth of the WASTCArD raw biopotential data (DC to 1 kHz), a preliminary digital filtering stage was applied to the Chest Lead I signal channel and to the three arm unipolar biopotential signal channels (Ch8, Ch9 and Ch10). This preliminary processing step was specifically configured to emulate the embedded ECG pre-filtering operation used in current wearable arm-band ECG monitoring device development [14,16] undergoing clinical trial. Therefore, in this study, only conventional pre-filtering was applied for ECG monitoring. This was implemented by first removing the DC offset, followed by digital 50 Hz notch filtering with a Q-factor of 20, then a digital 0.7 Hz high-pass, 2nd order Butterworth filter, and a 40 Hz low-pass, 4th order Butterworth digital filter, for reducing unwanted noise and produce a pr-filtered clearer signal for analysis.

### 2.3. Bipolar Arm Leads Definitions

Six bipolar arm leads as defined in Table 1, were derived from the three biopotential arm sensor channels referred to the GND electrode: Ch8, Ch9 and Ch10, by their linear combination or weighted sum equations as presented in Table 1 left column, and illustrated in Figure 1, by considering the perspective view of the Ch8, Ch9 and Ch10 skin-electrode contact points, forming a scalene triangle type as the BIS Quatro^TM^ sensor strip is wrapped around the arm, and also that the distal Ch8 electrode falls close to the 180° axis (or 0° axis), measured anticlockwise (arm top view as depicted in Figure 1), with respect to the chest plane and the proximal Ch10 point at the arm axilla.

The proposed six bipolar arm-leads (Table 1) are not only derived from a single unipolar arm-Lead; in a way as for the arm Lead-5 definition (Ch8 − GND), but also by the difference of two diametrically opposite unipolar arm-leads in the scalene triangle formation (Ch8 − Ch9 − Ch10), and also some bipolar arm-lead definitions were derived as the difference of a selected unipolar arm-lead and a weighted mean value combination of arm unipolar leads (e.g., arm Lead-4), aiming to approximately form an electrocardiographic vectorial axis orientation at a particular angle of interest. Thus, the six equiangular spaced bipolar arm-leads were defined for their ECG radial axis orientation analysis. For instance, bipolar Lead-2 (in Table 1) is the difference vector of electrode Ch10 with the mean of the other two electrodes (Ch9 and Ch8), in a similar way that standard “augmented limb leads” are defined, meaning that the defined bipolar arm-Lead forms a radial axis line between the Ch10 electrode and a projected midpoint of Ch8 and Ch9 electrodes as illustrated in Figure 1.

Therefore, the linear combination definition equations used for the proposed six experimental bipolar arm-leads were derived based on criteria for generating approximate equiangularly spaced electrocardiographic interelectrode axes vectorial orientations at 30° rotational steps, starting with Ch10 near the axilla, as the 0° axis reference point and parallel (co-planar) to the torso chest plane. The bipolar arm **Lead-1** (Ch8 − Ch10) axis is parallel to the chest plane forming a 0° angle, arm **Lead-2** [Ch10 − {(Ch8 + Ch9)/2}] has the axis rotation approximately about 30°, and the arm **Lead-5** (Ch8 − GND), can be seen as a negative rotation axis, forming a similar but negative angle; which is at about −30°. In a similar way, **Lead-3** [Ch9 − {(Ch8 + Ch10)/2}] and **Lead-6** (Ch9 − Ch10) also portray equal and opposite axes angles, approximately at −60° and 60°, respectively, while **Lead-4** [Ch9 − {(Ch8 + Ch10)/2}] axis falls approximately perpendicular to the torso chest plane, forming a −90° angle as presented in Figure 1. Nevertheless, these axes rotational angles are approximate due to the BIS Quatro^TM^ fixed sensor strip length and the actual subject’s arm diameter (or MUAC), which slightly varies from patient to patient in the selected narrow dataset of 18 subject cases presenting close BMI values within ±2% tolerance. From the arm circumference (MUAC) mean value and SD figures in Table 2, the derived arm diameter values (MUAC/π) are: mean value of 9.1 cm and SD of 0.16 cm.

### 2.4. Data Processing Approach

The ECG signal recorded from the upper left arm has a low amplitude (about 70 µV p-p) and is corrupted with various noise types from movement artefact, muscle generated electromyographic noise and mains interference (50 Hz/60 Hz). Conventional signal averaged ECG (SAECG) processing techniques, such as heartbeat alignment by cross-correlation with a pre-defined template beat, also known as maximum coherence matching, is widely accepted [23,24] and commonly used in SAECG analysis. However, the conventional SAECG approach is not advantageous for investigating clinical quality signal processing (denoising) in this unconventional arm-ECG study [18,23]. Thus, a robust Single-Fiducial-Point (SFP) alignment technique was adopted for the SAECG process in this work; as adopted in previously reported works [7,12,15,17]; the SFP technique provides a single and highly deterministic time point with respect the ventricular depolarisation event in the ECG, derived from processing the stable Chest Lead I signal as the reference QRS event timing, thus permitting an accurate arm-ECG beat alignment process. Further analysis of arm leads QRS detection adopted the conventional Pan-Tompkins (PT) QRS detection algorithm [19,25,26,27]. It offers better results for evaluating bipolar arm lead sensitivity, precision (positive predictivity) QRS detection and HRV metrics performance attributes; when referred to true positive QRS events detected on the Chest Lead I ECG, which can be taken as the absolute true event annotation [27]. Both of these approaches (SFP and PT) were implemented using embedded signal adaptive processes for moderately fast (within 6 heart beats) appropriate adjustment (adaptation) of amplitude and time thresholding ECG parameters [26].

#### 2.4.1. Signal-Averaged ECG Algorithm

For extracting the ECG QRS deterministic time alignment point of each incoming heartbeat, the selected pre-filtered ECG reference signal (the Chest Lead I in this study) is first bandpass filtered at 3–30 Hz (SFP filtered) [24]. Both the amplitude and time thresholding are made dynamic and adaptive to the ECG signal variations following simple rules of adaptation. The time thresholding algorithm in the SFP processing was designed to allow a 20% tolerance margin on the R-wave-like event pulse width thresholding criteria. For this study, the accurate SFP extraction time series in the reference Chest Lead I ECG signal, was utilised directly for 700 ms heartbeat centred (P-QRS-T frame) signal windows alignment in the averaging process for each bipolar arm-Lead ECG signal. Thus, the signal averaging algorithm generated a 700 ms SAECG frame (see Figure 2) around each SFP time mark, spanning from 320 ms before and 380 ms after the occurrence of the SFP time alignment mark. These PQRST signal frames were then accumulated (summed at every incoming heartbeat) and at the end, it is divided by the number of accumulated valid beats to produce the output signal-averaged 700 ms frame (SAECG frame), as illustrated in Figure 2.

#### 2.4.2. Bipolar Arm-Lead ECG Event Discovery Quality Assessment Metrics

The event discovery quality (EDQ) assessment of each of the six bipolar arm-Lead ECG signals was approached by calculating ratios of measured standard deviation (SD) value of waveforms within defined ECG segments. The QRS event signal-to-noise ratio (SNR), the P-signal-to-QRS ratio (PqrsR) and the T-signal-to-QRS ratio (TqrsR) were determined on the 700 ms SAECG frame around the SFP point (timing mark) fixed at 320 ms of the frame time scale. Thus, the 700 ms window is locked around the SFP point in a way that there are 320 ms to the left and 380 ms to the right of SFP point. The ratio calculations were performed by defining a primary QRS signal window that spans 160 ms centred around 40 ms away before the SFP time mark, thus, it is centred at 280 ms away from the beginning of the 700 ms window (see Figure 2), and a noise window 40 ms wide, centred at 40 ms from the left side border of the 700 ms SAECG frame. The windows for P-signal (100 ms) and T-signal (300 ms) SAECG signals were positioned adjacent to the primary QRS signal window, on the left side (contiguous) and on the far right side (at the end of the frame), respectively, as illustrated in Figure 2. For the SNR measurement the standard deviation (SD) of the SAECG signal (QRS) window (160 ms; Figure 2) was divided by the SD of the background noise signal in the noise window (Equation (4)). Higher values of the SNR are associated with better QRS signal quality. The PqrsR (Equation (5)) and the TqrsR (Equation (6)) ratios were calculated as [%] values. The PqrsR ratio [(P-signal/QRS) × 100], and the TqrsR ratio, [(T-signal/QRS) × 100], were calculated by computing the SD of the P-wave SAECG signal in its window width of 100 ms, and the SD of the T-wave SAECG signal in its window width of 300 ms, and then, respectively, dividing each of them by the SD over the SAECG QRS signal window width of 160 ms, and then multiplied by 100 for conveniently express the ratio as [%] values.
(4)$$\;\mathrm{SNR}=\frac{\mathrm{SD}\;\mathrm{of}\;\mathrm{QRS}\;\mathrm{Signal}}{\mathrm{SD}\;\mathrm{of}\;\mathrm{Noise}\;}$$
(5)$$\;\mathrm{PqrsR}=\frac{\mathrm{SD}\;\mathrm{of}\;P\;\mathrm{Signal}}{\mathrm{SD}\;\mathrm{of}\;\mathrm{QRS}\;\mathrm{Signal}}\;\times \;100$$
(6)$$\;\mathrm{TqrsR}=\frac{\mathrm{SD}\;\mathrm{of}\;T\;\mathrm{Signal}}{\mathrm{SD}\;\mathrm{of}\;\mathrm{QRS}\;\mathrm{Signal}}\;\times \;100$$

#### 2.4.3. QRS Event Detection Performance

The Pan-Tompkins (PT) algorithm uses the slope, amplitude, and width of an integrated window for the identification of QRS complex events in the ECG [19,25]. Amplitude and pulse width thresholding parameters in the PT algorithm decision stage were set specifically for discriminating and validating QRS complexes of sinus beats; rejecting potential noise peaks in the algorithm output, reducing the false events detection rate produced by non-ECG signal components and artefacts present in the Chest Lead I and arm-ECG signals [26,27]. To adapt to slow variations in QRS amplitude and duration, respective threshold variables were designed to adjust automatically at every positive decision stage (valid QRS event detection).

#### 2.4.4. Arm Leads QRS Detection Performance Analysis

The arm-ECG QRS detection performance of prior defined bipolar arm-ECG leads (Table 1), was analysed against the reference Chest ECG Pan-Tompkins detected QRS events serving as the annotated absolute true positive instances, for the sensitivity (*Se*)% and positive predictive value (*PPV*)% or detection precision, as the main QRS detection performance assessment parameters. These were calculated using the formulas in Equation (7) and in Equation (8), respectively, computed after determining the number of true positive (*TP*), false negative (*FN*) and false positive (*FP*) QRS events on the arm-ECG leads as detected by the above-described *PT* algorithm [25] on the arm-ECGs. A dedicated algorithm which used the valid detected Chest Lead I QRS events with the *PT* process as absolute true occurring events, generated the *TP*, *FN* and *FP* counts upon independently detected QRS events on each of the six defined arm-leads, as output by the *PT* QRS detection processes on the arm-Lead ECGs.
(7)$$Se=\frac{\mathrm{TP}}{TP+FN}\;\times \;100\;(\%)$$
(8)$$PPV=\frac{\mathrm{TP}}{TP+FP}\;\times \;100\;(\%)$$

In a *PT* method testing tool, the true positive (*TP*), false negative (*FN*), and false positive (*FP*) QRS events on the arm leads ECGT were identified by a dedicated algorithm. Each arm-lead QRS detection was checked with the reference ECG *TP* event annotations provided by the Chest Lead I. Arm-Lead QRS detection simultaneous agreement with the annotated *TP* event from the Chest-lead data is considered true positive detection on the arm-Lead. The QRS that were not detected on the arm ECG but are present on the Chest are the false negatives. The QRS-like peaks which are detected on the arm lead but actually do not appear in the Chest reference signal are the false positives.

Another useful metric for arm leads QRS detection performance analysis considered in this study is by simple evaluation of the correlation between the Number of arm-Lead TPs (NTPqrs) and the Number of Valid Chest-lead QRS (NVChestQRS) events for the 18 subject cases data points in a scatter plot. The Pearson correlation coefficient (p) [28] and the scatter plot associated trendline Coefficient of Determination (R^2^) were used as the assessment performance metrics in this aspect.

#### 2.4.5. HRV Performance Metrics

Heart rate variability (HRV) is a clinically important and prominent CVD diagnostic factor. Inter-beat intervals are the time intervals between successive QRS complexes originated by sinus node event in a continuous long ECG recording [4]. Since HRV is a highly individualised measure, long-term continuous ECG and HRV tracking using a non-invasive armband-based wearable monitoring device is an appealing option for HRV trend-based indicator of general health. It’s also a promising instrument for assessing correlation between arm-ECG leads HRV measurements and their corresponding standard measurements on the Chest Lead I ECG signal. Widely used HRV time-domain indices [5] usually reflect the variations in inter-beat interval (IBI) between consecutive heartbeats in milliseconds (ms). A simple time-domain HRV metric, regularly used in comparative studies [6], is the root mean square (RMS) of the daily heartbeat intervals (beat-to-beat); it reflects the dynamic fluctuation range in heart rhythm [4].

For the arm-ECG leads axes orientation analysis around the MUAC line, the HRV RMS metric was measured independently (standalone basis) on the six arm leads, for each of the 18 subject cases, and correlated with the HRV RMS value measured on the Chest Lead I, for the 18 cases data points, using the conventional Pearson correlation coefficient (p) [28], and also the scatter plot trendline Coefficient of Determination (R^2^), for comparative HRV metric performance presented by each of the six bipolar arm-ECG leads with orientation axes 30° stepwise around arm. The HRV RMS metric was measured on 8-min-long ECGs of the 18 cases. For this branch of study, the conventional Pan Tompkins algorithm was implemented autonomously and independently for Chest Lead I QRS-detection and for Arm-ECG QRS-detection. In the latter process, estimation of the IBIs was implemented for dealing with missing beats.

### 2.5. Removing Outliers

The resulting data from each of the above performance processing methods: SAECG based ECG event discovery quality (ratios in % figures), PT QRS detection and HRV-metrics correlation (arm-measurement vs. chest-measurement) were further processed systematically (following statistical methods) in order to eliminate outlier values. Statistically, outliers are the data points that differ significantly from other observations. These may cause distortion and skewed data leading to inaccurate analysis of results. For the exclusion of such values the SNR, PqrsR, TqrsR, Se, PPV, NTPs, SDNN, RMS and IQRNN output data vectors were further processed in MATLAB using the *rmoutliers* command [29]. Basically, this MATLAB statistical operation command removes the elements which are more than three scaled median absolute deviation (MAD) from the median of the data vector [30,31]. The MAD is a statistical function that measures the degree of dispersion of data, i.e., it determines in a data set, by how many values it is likely to differ from the mean [30]. This operation enabled the systematic removal of any of the extreme values from the results of performance assessment parameter values. Furthermore, after inspection of the *rmoutliers* command output data in all cases, the process was found to be reasonable and fair, hence, was considered reliable as an objective data handling tool particularly suitable for this study.

## 3. Results

### 3.1. SAECG Bipolar Arm Lead Signal Enhancement Output

The ECG signal averaging process, using the Chest Lead I for finding the QRS deterministic feature (SFP) time mark at an accurate position in the ECG, taken as the beat alignment time reference for the SAECG process on the six bipolar arm leads, was tested and validated. This arm SAECG approach effectively enhanced the arm-ECG signal quality for the study. It enabled the comparative assessment of bipolar arm leads event discovery performance over baseline noise level, or relative to ECG QRS event signal presence level and quality in the same arm-lead. The signal averaging alignment time mark for producing the P-QRS-T frame, for every incoming validated beat, was derived from the reference Chest Lead I QRS events time marks sequence. This achieved effective denoising operation by SAECG on the six arm leads SAECGs and on the Chest ECG itself, which can only be possible by an accurate P-QRS-T frame alignment process. Figure 3 illustrates the resulting 700 ms SAECG frames (PQRST) of a representative subject case, after averaging 564 valid heartbeat frames. ECG event discovery quality (EDQ) on each of the six arm-leads can be appreciated in this particular subject. Of particular interest is the P-wave signal discovery on the six bipolar arm leads; particularly on arm Lead-3, arm Lead-4 and arm Lead-6, as corroborated by the PqrsR mean values shown in Table 3, thus, illustrating the effectiveness of the adopted arm-SAECG method in this study.

### 3.2. Bipolar Arm Leads ECG Waveform Events Discovery Quality (EDQ) Analysis

The QRS signal to noise ratio (SNR), P-signal-to-QRS ratio(%) (PqrsR), and T-signal-to-QRS ratio(%) (TqrsR) metrics provided comparative ECG event signal strength assessment of ECG waveform component events within a particular ECG lead, or in comparison with other leads, for the EDQ study branch over the 18 subject cases. Table 3 summarises the study branch-1 mean and standard deviation (SD), median and interquartile range (IQR) values computed after applying outliers removal operation in order to improve normal data distribution characteristics, for the SNR, PqrsR and TqrsR metrics on the six arm leads having systematic arm-Lead radial axis rotation positions around the arm MUAC line, and on the Chest Lead I as comparative contrast (1st row in Table 3). It can be observed from Table 3 and from its convenient graphical bar chart presentation in Figure 4 (summarising only mean and median values), that arm Lead-1 (axis at 0°), arm Lead-2 (axis at 30°) and arm Lead-5 (axis at −30°) resulted in the highest median values for QRS events SNR: 144, 145 and 271, respectively. With regard to ECG T-wave event discovery; the 2nd largest ECG event in the P-QRS-T signal components, arm Lead-1 (axis at 0°), arm Lead-2 (axis at 30°), arm Lead-5 (axis at −30°) and arm Lead-6 (axis at 60°) presented the highest median values of TqrsR: 35.1%, 41.9%, 37.0% and 33.9%, respectively. Nevertheless, it is evident the relatively high Interquartile Range (IQR) and SD values throughout the metrics results presented in Table 3; revealing a large scattering nature of data points in the particular EDQ performance metrics study results summarized in Table 3, even after having removed outlier cases, as described in Methods Section 2.5, and may render these particular metric results being prone to some uncertainty, as there is considerable natural variations in each subject’s ECG, particularly on the T-wave to QRS ratio (TqrsR) values, and patients’ broad demographic characteristics (in Table 2), except for BMI values, which have been selectively kept within a very narrow range, centred around 24.9 (Kg/m^2^). Thus, care must be taken when interpreting the EDQ performance metrics results presented in Table 3.

Overall, it is interesting to note that the axes for QRS discovery are angularly adjacent, thus indicating that arm QRS events are best observed in bipolar arm electrodes axis positioned in the angular fan between 30° and the −30° axes. In a similar perspective from Table 3 and Figure 4, the TqrsR [%] EDQ metric median values in Table 3 indicate that arm ECG T-waves are best observed in bipolar arm electrodes axis positioned in the angular sector between −30° and the 60°, which is within the wider clique of arm-Lead radial axis orientation for T-waves recording, incorporating Lead-1, Lead-2, Lead-5 and Lead-6, hence, overlapping the clique of arm-Lead radial axis orientation for QRS events recording (Lead-1, Lead-2 and Lead-5). In contrast, arm Lead-3 and Lead-4 are not advantageous for recording ECG T-wave events on bipolar arm leads.

In another perspective, the mean PqrsR [%] results in Table 3 are lower than TqrsR (%), as expected, and having maximum values of 8.35% for arm Lead-3 (axis at −60°), 7.79% for arm Lead-4 (axis at −90°; perpendicular to the chest plane) and 7.03% for arm Lead-6 (axis at 60°; an axis which is also at −120° in its extended version and is adjacent to axis at −90°). Again, it is interesting to note that these three arm-leads axes are angularly adjacent and thus form an axis angular fan between −60° and −120°, thus, forming a clique of the arm-Lead radial axis orientation for somehow advantageous tendency for P-wave recording, incorporating Lead-3, Lead-4 and Lead-6. These relative signal strength estimation results are valuable for this particular ECG event waveform component and thus provides important P-wave discovery quality metrics for bipolar far-field arm-leads. Additionally, the latter results reveal, to some extent, that orthogonality between ventricular activity (QRS-T events) and atrial activity (P-wave) is preserved in upper-arm bipolar leads.

### 3.3. Arm Leads QRS Detection Sensitivity (Se) and Precision (PPV) Performance Analysis

Bipolar arm-Lead far-field ECG signals are expected to be of relatively low-amplitude around 70 µV p-p, and are expected to be considerably contaminated with unwanted electromyographic activity. Nevertheless, with adequate ECG denoising methods [14] and optimal bipolar arm electrode pair positioning, the arm-ECG can extract a clinically acceptable ECG signal. With this in mind, in this 2nd branch of study, a more clinically oriented use of arm-ECG long-term monitoring complementary performance criterion for selecting the most likely favourable arm electrode positioning in a wearable armband device was defined. Thus, evidence-based knowledge on the PT based QRS events detection performance assessment of the six arm leads under consideration would be of important value.

Table 4 and bar charts in Figure 5 present result values obtained for the sensitivity (Se%) performance metric; which indicates the true positive rate, and for the positive predictive value (PPV%) performance metric, which indicates the QRS detection precision. The mean, SD, median and IQR results in Table 4 can reveal the QRS detection performance of the six arm-leads, based on the Pan-Tompkins approach described in Section 2.4.4. It can be observed that, consistently with the results in Table 3, arm Lead-1 (axis at 0°), arm Lead-2 (axis at 30°) and arm Lead-5 (axis at −30°) resulted with the highest Se% metric median values: 93.3%, 85.1% and 92.2%, respectively, and also these three arm leads presented the highest median values for detection precision, PPV (%): 99.6%, 99.6% and 99.3%, respectively. Once again, it is interesting to note that these axes are angularly adjacent around the 0° axis (coplanar to chest plane) and corroborate the arm leads QRS events clique of arm leads.

As an intuitive and ease of clear interpretation analysis on the arm leads QRS detection performance in study branch-2, the simple evaluation of the Pearson correlation coefficient (p) [28] between the total Number of arm-Lead TP events (valid arm ECG QRS events as annotated by the Chest Lead I, in the 8 min ECG recording, NTPqrs) and the total number of Chest Lead I detected Valid QRS events (NVChestQRS), according to the PT algorithm, and it’s set of QRS event time and amplitude thresholding parameters, for reliable event validation, as conveniently presented graphically as a bar chart in Figure 6 for the 18 subject cases data points, and in a scatter plot for each of the six arm leads; as presented in Figure 7. Both the *p* value and the R^2^ value results for this complementary analysis are summarised in Table 5.

Results summary of Pearson correlation coefficient (p) and scatter plot (Figure 7) associated trendline coefficient of determination (R^2^) between HRV RMS metric (of sinus beats IBI) measured on each of the six bipolar arm leads and on the Chest Lead I ECG (as the gold standard).

From the results of this QRS event detection and NTPqrs vs. NVChestQRS correlation performance study presented in Table 5 and in Figure 6, for the six arm leads, it can be comparatively observed that arm Lead-1, Lead-2 and Lead-5 presented the highest Pearson correlation (p) values: 0.88, 0.86 and 0.83, respectively, and also the highest scatter plot trendline Coefficient of Determination (R^2^) values: 0.77, 0.74 and 0.68; and particularly revealing Lead-1 (axis at 0°) as the top performing arm-Lead for the arm-ECG QRS detection NTPqrs vs. NVChestQRS correlation metrics.

### 3.4. Arm Leads Comparative HRV Metrics Measurement Performance Analysis

The HRV metrics results are presented in Table 6, in Figure 8 bar charts and in Figure 9 scatter plots. From the results summarised in Table 6 and Figure 8 on this particular QRS event detection assessment for measuring HRV RMS metric, on 8 min ECGs from the six defined bipolar arm leads in this 3rd-branch of study, which correlates the HRV RMS metric measured conventionally from the Chest Lead I ECG, with the simultaneously recorded arm-ECG leads, in the same subject case, as described in the Methods Section 2.4.5.

In this 3rd-branch of study, arm Lead-1, Lead-2, Lead-5 and Lead-6 presented distinctively the highest correlation figures for the HRV RMS metric against the gold standard values measured in the Chest-Lead, both in p and R^2^ coefficients values: *p* = 0.97 and R^2^ = 0.95, *p* = 0.96 and R^2^ = 0.93, *p* = 0.99 and R^2^ = 0.98, *p* = 0.97 and R^2^ = 0.93, respectively, as can be appreciated from Table 6. In contrast, arm Lead-3 and Lead-4 consistently presented the lowest HRV RMS correlation with Chest RMS values: *p* = 0.26 and R^2^ = 0.07, *p* = 0.89 and R^2^ = 0.80, respectively. Thus, these results are in agreement with the indications from the other study branches on best arm leads on QRS events discovery tendencies.

Therefore, with regard to arm leads HRV monitoring potential, on a standalone basis, using the conventional PT QRS detection algorithm, the associated best performing bipolar arm electrodes pair radial axis orientation angular range fanout presenting reliable HRV RMS measurement values quite close to values measured from the standard Chest Lead, greatly correspond to the same axial angular range defined by the main clique of arm leads: Lead-1, Lead-2 and Lead-5, according to the three study branches on arm-ECG ventricular activation QRS complex waveforms signal quality and reliable processing for potential clinical use.

### 3.5. Overall Results Summary

For summarising and to consolidate the interpretation of the diverse analysis results delivered by the three branches of analysis in this study, it is helpful to refer to the four bar charts in Figure 5, Figure 6 and Figure 8, as these charts graphically reveal the main supporting results data for deriving the following Figure 10. The four bar charts easily depict which arm leads do present relatively poor performance for recording and examining certain ECG waveforms in the arm-ECG. The latter important observation conforms part of the key results of this study yielding the diagram in Figure 10, which highlights the arm lead axes angular sectors offering best discovery tendency for particular arm-ECG events waveform (QRS complex, T-waver and P-wave) recording, detection and analysis. Thus, Figure 10 summarises in a handy rough map diagram the integrated key results conclusion on bipolar arm-lead electrodes pair axis angular orientation around the MUAC line. Thus, three main axes angular position ranges or fans are distinguished; interestingly, the larger angular fan associated with ventricular activity (QRS-T, blue and green shaded) and the smaller atrial activity related (P-wave, beige shaded) axis angular fan are interrupted by a 30° gap from the ventricular activation (QRS-complex, blue shaded) vector axis angular sector; the 30° arm ECG quiet (blank) angular sector is between a QRS event low performing arm-Lead radial axis orientation: arm Lead-3, and a P-wave low EDQ performing arm-Lead axis: Lead-5. Furthermore, to some extent, Figure 10 reveals that the orthogonality between ventricular activation (blue shaded angular fan) and atrial activation (beige shaded angular fan) is preserved in the upper-arm bipolar leads. This particular finding is of clinical importance for the future clinical practice with ECG armband sensors. Additionally, the overall results of this study reveal that each ECG event waveform has three (for QRS event and for P-wave) or four (for T-wave) relatively best performing bipolar leads which delimit the range of axis angles that are likely to be most suitable for the recording, detection and analysis of the electrophysiological event of interest on the arm.

According to the favourable tendency revealed by three adjacent arm leads: Lead-1, Lead-2 and Lead-5 in this study, QRS complexes are best discovered on the arm-ECG when the arm-lead radial axis orientation is in the angular fan from −30° to 30°, including the 0° axis (refer to Figure 10). ECG P-wave signals are best discovered on the arm when the arm-lead axis is within the angular fan from −60° to 60° (equivalent to −120°), including the −90° axis, as relatively enhanced results are observed with adjacent arm Leads 3, 4 and 6. Finally, as ECG T-wave event signals were best discovered with four adjacent arm leads: Lead-1, Lead-2, Lead-5 and Lead-6, this revealed that the axis angle ranging from −30° to 60° would be the axis angular fan for favourable arm-ECG T-wave discovery. All these results remarks are summarised in Figure 10.

## 4. Discussion

The results from this study have provided new insights for understanding the ECG characteristics of far-field bipolar ECG leads from the left upper arm, which offers an attractive anatomical location for non-invasive long-term ECG monitoring armband devices. Thus, this research work has addressed the current knowledge gap relating to the best position of ECG bipolar sensor electrode pairs around the MUAC line. Here, six derived equiangularly spaced axes (at 30° steps) of bipolar arm-leads sensor pairs around the armband were analysed based on clinical data.

In the first branch of the study on Event Discovery Quality (EDQ) performance, the six defined arm-leads were assessed on their ability to record ECG by signal strength ratio metrics: SNR (QRS), PqrsR and TqrsR, which were with respect to background noise or largest deterministic signal event within the heartbeat related P-QRS-T time period in the ECG. In the second branch of the study, detection performance of the main ECG waveform component (QRS-complex) using a conventional algorithm (Pan-Tompkins), was investigated in the six arm-lead axes of vectorial orientations around the arm. Then, in the third branch of study, the more challenging HRV analysis but clinically relevant metric, was evaluated (measured) in the 8 min ECGs duration using the armband bipolar ECG lead method and correlated with the standard measurement method from a limb Lead ECG, and the resulting correlation metrics (p) and (R^2^) used as performance stratification of arm-Lead axes on HRV measurement. The overall results obtained from the three study branches of pilot study are very encouraging for considering the prospective clinical use in certified medical field and eventual commercialisation of a final armband arrhythmia monitoring device.

The EDQ results revealed by the ECG signal averaging process with the P-QRS-T frames alignment reference point, extracted from the reference Einthoven’s Lead I (from the Chest) QRS events, provided accurate signal averaged frames alignment which enabled an effective SAECG denoising operation on the six arm-lead ECGs, as well as on the Chest ECG itself, as illustrated in the set of P-QRS-T SAECG frames in Figure 3. There, the particular subject case P-wave event is clearly discernible within the 100 ms allocated signal window adjacent to the left of the QRS signal window (160 ms) in the resulting SAECG P-QRS-T frame (Figure 2). This approach resulted in the remarkable PqrsR (%) statistical values presented by arm-leads 3, 4 and 6, in Table 3 and Figure 4. From the SNR metrics results in the same table and figure, these three arm-leads would have been regarded as relatively poorly performing arm-leads and just dismissed as of little use. Therefore, possible optimal bipolar arm-Lead positioning depends on the particular ECG event signal of interest, thus, necessitating at least two bipolar arm-leads at axes in line with arm Lead-1 for QRS and T-wave events discovery and the other pair of electrodes axis in line with arm Lead-4 for P-wave event discovery on the arm-ECG.

From the results observations derived from Table 3, Table 4, Table 5 and Table 6 from the bar charts in Figure 4, Figure 5, Figure 6, Figure 7, Figure 8 and Figure 9, and scatter plots presented in Figure 7 and Figure 9, it is evident the wide dispersion of data points in this study, even after outliers removal operation. This is particularly true for the QRS SNR related IQR and SD figures in all the arm-leads in Table 3. This experimental fact is to be expected as there is considerable variation in subjects ECG waveform amplitude, pattern and background noise, and data normal distribution is not expected in this relatively small study. Thus, the results of this pilot study need to be handled and interpreted with due care. Some arm leads performance metrics (e.g., median of PqrsR in Figure 4) did not yield to clear differentiation for arm leads performance stratification. Nevertheless, in the 2nd study branch, regarding the QRS detection precision performance (PPV) in Table 4, the related IQR and SD figures are remarkably low.

The Pan-Tompkins algorithm was used to evaluate the detection of QRS complexes in standard chest ECG Lead I compared with true positive QRS events in arm ECG signals, with reference to valid QRS annotations in the Chest Lead I, and also to generate the useful and intuitive correlation analysis (p and R^2^) presented in Table 5 and scatter plots in Figure 7 with added information of R^2^ values of the trendlines, which provided a supportive stratification parameter for arm leads axes performance around the arm. The implemented PT algorithm was capable of performing the QRS detection process for both high-quality chest and noisy arm ECG data in a standalone mode. The Sensitivity (Se) and Positive predictive value (PPV) calculated for the six bipolar leads clearly reveal which arm leads have relatively weak QRS detection performances; if decision threshold values criteria is to be set, then, it could be set low performing arm leads as have median values of Se% and PPV% below 80% and 99%, respectively. In contrast, it was revealed that the adjacent arm Lead-1, Lead-2 and Lead-5 performed the best of the six arm-leads proposed for this study; with the highest Se median values above 93% and PPV values above 99%. These indicators are easily appreciated in the bar charts of Figure 5, with amplified scale ranges (zoom-in) to facilitate depicting performance differentiation between arm leads.

The results from this study may be contrasted with the findings in a previous study carried out on a pilot patient database (N = 11) [27]. There, prior to QRS detection performance analysis with the PT algorithm, arm-ECG signal denoising was successfully achieved by an advanced Discrete Wavelet Transforms (DWT) ECG filtering technique using a 4th-order wavelet decomposition, with the db4 Daubechies wavelet as the mother wavelet. Then, the performance analysis results [27] indicated that the bipolar lead on the upper-arm, defined as Ch8 − Ch10, presented the best sensitivity mean value (99.8%) and mean PPV (98.4%) performance. In contrast, the performance results obtained here indicated a maximum sensitivity mean value of 86.8% (with arm Lead-1). This lower figure was expected due to the use of a basic and conventional pre-filtering process in order to emulate the embedded real-time pre-filtering process employed in the current wearable arm-ECG device prototype undergoing clinical trial [14,16]. Therefore, in general practice it is recommended to always incorporate some advanced data-driven pre-denoising filter technique, such as a DWT based one [32], as was previously investigated by the team in the present study; in order to significantly improve the arm-ECG QRS-detection sensitivity performance value to a more competitive standard level.

In a different perspective, from a more recent pilot study investigation using a small dataset (N = 11), with similar objective of finding the best upper-arm bipolar lead axis orientation was investigated without using the gold standard ECG signal averaging denoising method [16]. There, under such rudimentary signal pre-conditioning, the bipolar lead, defined as Ch8 − Ch10, was found to be the lead with the highest SNR mean value, with a corresponding axis angle orientation defined as −30°. In contrast, here, an arm-ECG signal averaging approach using a beat alignment technique referred to the Chest Lead-I, further investigated the proposed objective on discovering arm-ECG QRS events. Thus, the best results of arm-Lead mean SNR of QRS complexes obtained in this extended study; after performing ECG signal-averaging denoising, were significantly improved here: from 15.8 [16] to 244 (Table 3). Additionally, arm-ECG P- wave and T- wave signals discoveries were enabled by the effective SAECG denoising technique in this extended study using an SAECG denoising technique. This study, with a larger database of 30 subjects confirms that the bipolar lead Ch8 − Ch10, labelled here as Lead-1, and Lead-5 (axis angle at −30°) were found to be the best two bipolar leads for QRS detection, thus, corroborating the results obtained in the previous pilot study, with optimal adjacent axis angles findings presented here.

Regarding the summarising rough map diagram presented in Figure 10, it is important to broadly interpret the axis or radial vector lines in the diagram. As illustrated there, the angular fans for the three arm ECG events are considered: QRS-complex, T-wave and P-wave, as well as their respective delimiting axis lines, the sectorial fans validly project to the diametrically opposite side, with the two delimiting axis lines intersecting, forming a vertex with corresponding vertically opposite 30° angles, at a central geometric point which is best taken as the geometric centre of the arm circumference, and taking the 0° axis as being parallel to the chest plane at the same time that it passes through the said arm geometric centre.

This pilot study focused on the baseline characterisation of bipolar arm-ECG leads positioning under quiet conditions (subjects sitting at rest); without factors of higher level of complexity, such as normal activity movements. Therefore, baseline bipolar arm-ECG leads were characterised, and feasibility best conditions and limitations were stablished under restricted activity conditions. With this arm-ECG baseline knowledge, the research can now move on to the next stage of complexity, with confidence of what to expect. Thus, from the results of this study under quiet conditions, we can now be confident that by means of advanced machine learning support and artificial intelligence (AI) methods, the more challenging stage with normal activity conditions can be approached and overcome. For example, a few minutes (say 8 min) of initial machine training period to capture the patient’s arm-ECG baseline characteristics vector, will enable the smart trained machine to successfully handle the recording of arm-ECG under normal activity conditions; for an acceptable and stable arm-ECG trace quality for clinical use.

## 5. Conclusions

A single bipolar arm lead (a pair of electrodes diametrically opposite) in a wearable armband device can rotate around the arm; then, the main question addressed in this study was simply the following one: at which angle of rotation will this ECG armband perform best for heart rhythm monitoring? The results from the methodological analysis in three study branches presented here, revealed very interesting arm-ECG evidences illustrated in Figure 3, rough arm-ECG waveform components discovery tendencies in Figure 10, and in other results tables. These results open the possible horizon for arm-ECG monitoring for arrhythmia detection, including AF, P-QRS-T features extraction, ventricular tachycardia and ventricular ectopic beats, just to mention a few. Furthermore, the six derived bipolar arm-lead cases can be interpreted as virtual circular displacement of a simple armband with a single bipolar lead. Thus, this study has virtually addressed the effect of circular displacement around the arm of the bipolar lead in an armband. From this perspective view, the results consistently indicate that the effect of rotational displacement of the arm-lead is not critical, that is, there is no arm-ECG signal total fading away as the armband is rotated around the arm. The gained knowledge contribution from this work, strengthens the confidence on the underlaying hypothesis of ECG armband heart rhythm monitoring feasible clinical use.

The bipolar arm Lead-1 [Ch8 − Ch10] axial orientation, at axis rotation angle of 0°, was identified to be the preferred bipolar electrode deployment for armband ECG monitoring devices, as this arm-lead axis orientation presented the highest median of QRS events SNR metrics value compared to the other 5 bipolar arm-lead axes. Additionally, arm Lead-1 presented both the highest QRS detection sensitivity metrics median value (93.3%), precision PPV median value (99.6%) and highest HRV RMS metric correlation figures (p) and (R^2^) with HRV RMS values as measured on the Chest-Lead: 0.97 and 0.95, respectively. In contrast, the worst performing arm leads on QRS discovery and detection were Lead-3 and Lead-4, as can be easily depicted from all result Figures.

A knowledge revealing conclusion drawn from this study is depicted in Figure 10. There, the angular range from −60° (arm Lead-3 axis), through −90° (arm Lead-4 axis), to −120° (arm Lead-6, back projected) axes vectorial orientation of sensor pairs, offered the best P-wave event discovery tendency in the arm-ECG. This results interpretation reveals evidence of the vectorial orthogonal relationship between the ventricular (QRS-complex) and the atrial (P-wave) activations, and that this orthogonality observed by cardiac electrophysiologists in conventional vectorcardiography is preserved and is observable with bipolar leads around the upper-arm. This is very encouraging for further prospective studies with a much larger number of subjects and longer recording times using an armband ECG sensor device.

## Figures and Tables

**Figure 2 sensors-22-07240-f002:**
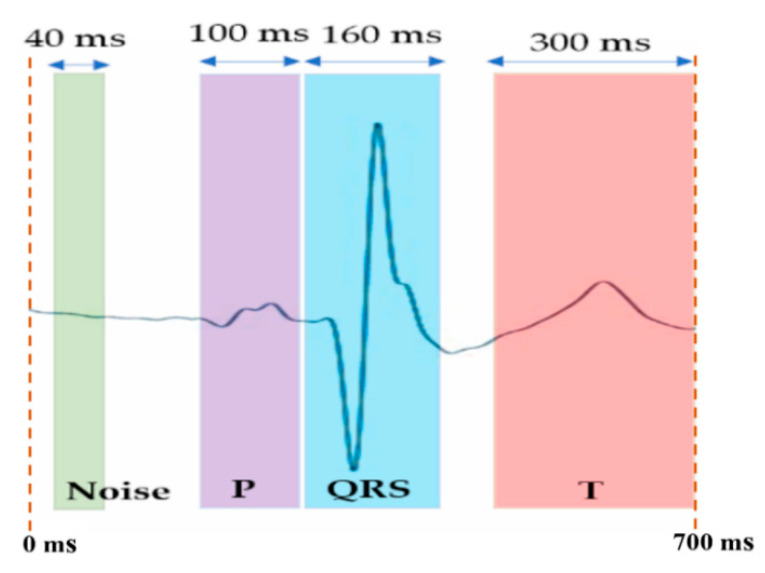
Noise and signal windows in the SAECG 700 ms frame (locked with respect the SFP point at 320 ms of the frame) for QRS signal-to-noise ratio (SNR), P/QRS ratio (PqrsR) and T/QRS ratio (PqrsR) analysis.

**Figure 3 sensors-22-07240-f003:**
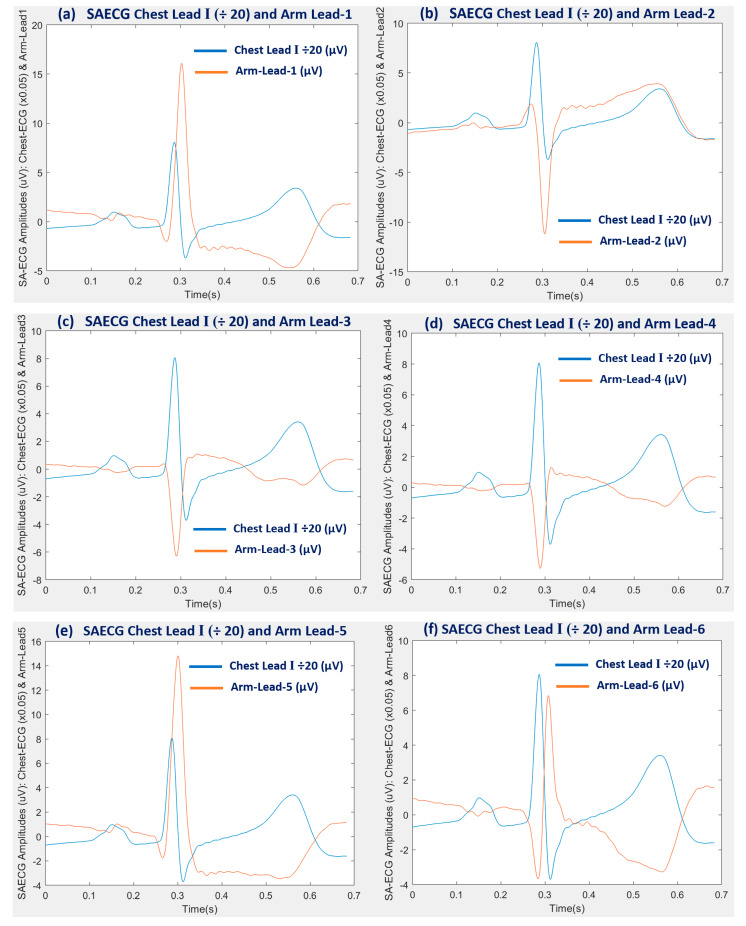
Example subject case results of signal averaged ECG denoising process P-QRS-T frames (700s), of reference Chest Lead I ECG signal (blue) and bipolar arm-ECG signals (brown) for the six arm leads (Lead-1 to Lead-6) under study, after averaging 564 validated heart beats.

**Figure 4 sensors-22-07240-f004:**
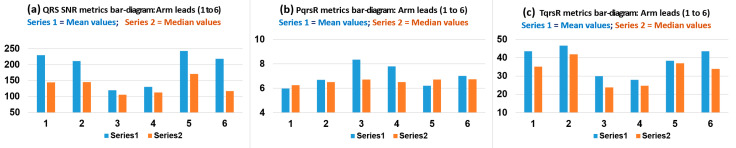
Bar charts presentation with amplified scale ranges (zoom-in) to facilitate depicting performance differentiation between the six arm leads mean and median value results (on the 30 subject cases) in Table 3: (**a**) for QRS SNR metric values, (**b**) for PqrsR (%) metric values and (**c**) for TqrsR (%) metric values.

**Figure 5 sensors-22-07240-f005:**
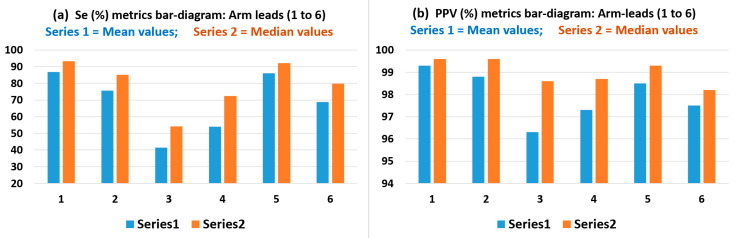
Bar charts presentation with amplified scale ranges (zoom-in) to facilitate depicting performance differentiation between arm leads of the Se (%) and PPV (%) mean and median value results in Table 4.

**Figure 6 sensors-22-07240-f006:**
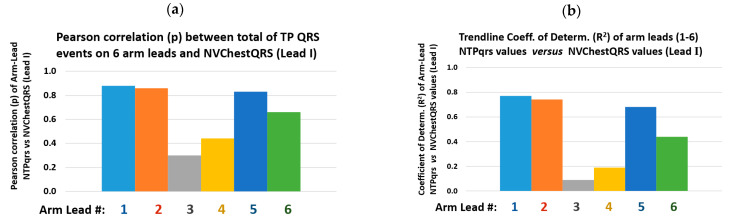
Bar charts for results in Table 5: arm-leads (1–6) values of NTPqrs vs. number of valid QRS in the Chest Lead I (NVChestQRS): (**a**) Pearson correlation (p), and (**b**) trendline coefficient of determination (R^2^).

**Figure 7 sensors-22-07240-f007:**
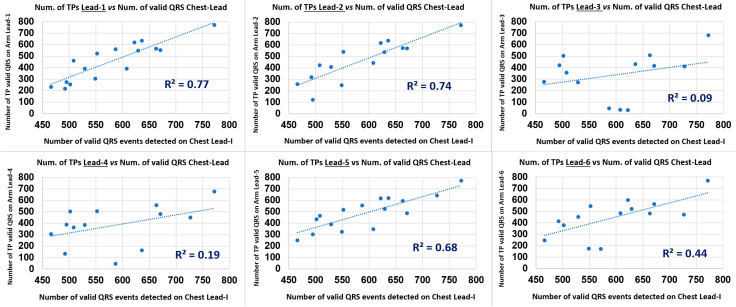
Scatter plot of 18 data pair/case points and associated trendline coefficient of determination (R^2^) of NTPqrs metric on arm leads (1–6) versus values of NVChestQRS on the Chest Lead-I.

**Figure 8 sensors-22-07240-f008:**
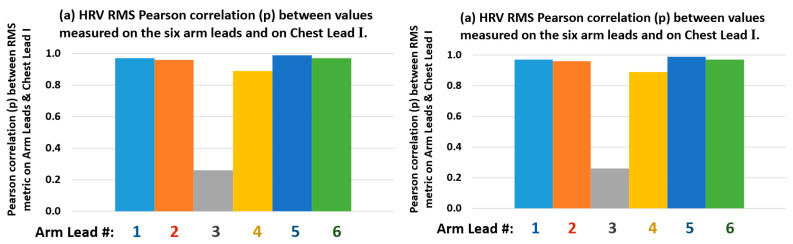
Bar charts for results in Table 6: arm leads (1–6) HRV RMS metric values versus measured HRV RMS values in the Chest Lead I: (**a**) Pearson correlation (p), and (**b**) trendline (Figure 9) coefficient of determination (R^2^).

**Figure 9 sensors-22-07240-f009:**
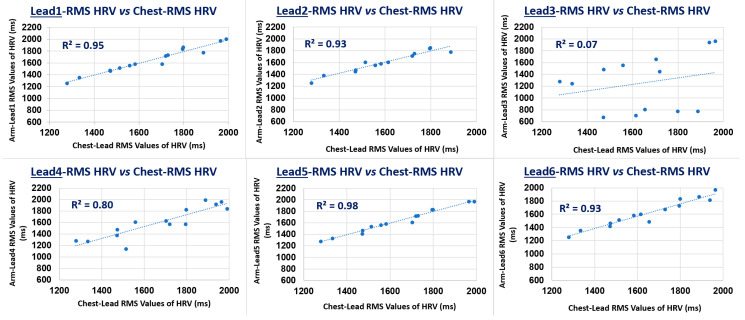
Scatter plot of 18 data pair/case points and associated trendline coefficient of determination (R2) of HRV RMS metric measured values (ms) on arm leads (1–6) versus measured values (ms) on the Chest Lead-I.

**Figure 10 sensors-22-07240-f010:**
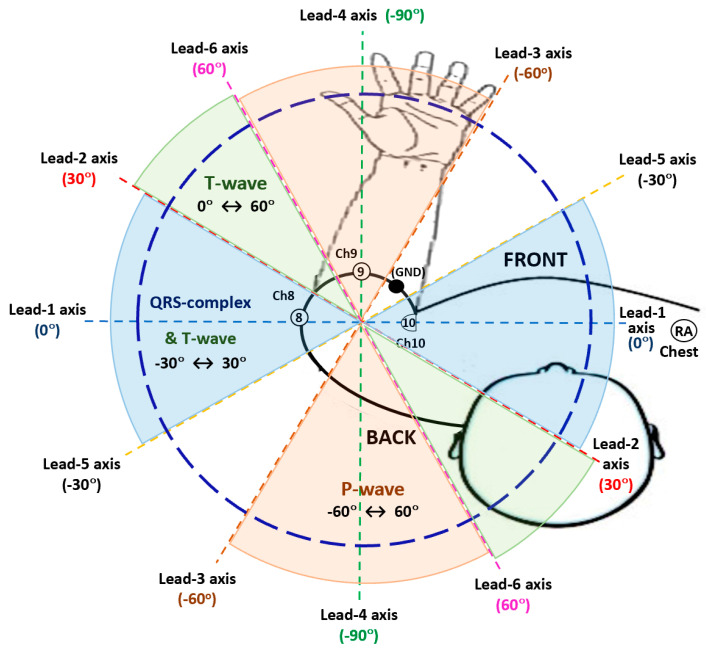
Bipolar arm-Lead axis angular range positions rough map for best recording of arm-ECG QRS-complex, T-wave and P-wave waveform components from an upper armband sensor (arm leads 1 to 6 are as derived from Ch8 to Ch10 and GND in Table 1).

**Table 1 sensors-22-07240-t001:** Bipolar Arm-Leads their definition and their reference axis rotation angles.

Bipolar Arm-Lead Definition	Bipolar Arm-Lead Axis Rotation Angle
** Lead-1: ** **(Ch8 − Ch10)**	**0°**
** Lead-2: ** **[Ch10 − {(Ch8 + Ch9)/2}]**	**30°**
** Lead-3: ** **[Ch9 − {(Ch8 + Ch10)/2}]**	**−60°**
** Lead-4: ** **[Ch9 − {(Ch8 + Ch9 + 2 × Ch10)/3}]**	**−90°**
** Lead-5: ** **(Ch8 − GND)**	**−30°**
** Lead-6: ** **(Ch9 − Ch10)**	**60°**

**Table 2 sensors-22-07240-t002:** Clinical database patients baseline demographics for included cases in this study (N = 18).

Characteristics	Mean	SD	Median	IQR
**Age (y): both genders**	40.3	15.3	37.0	21.8
**Age (y): females (77%)**	50.6	14.1	52.0	22.5
**Age (y): males (33%)**	27.3	4.7	29.0	4.5
**Height (m)**	1.68	0.12	1.68	0.17
**Weight (Kg)**	70.9	10.2	69.0	14.1
**BMI (** **Kg/m^2^** **)**	24.9	0.43	24.9	0.66
**MUAC (cm)**	28.6	0.50	28.5	0.76
**Pulse (bpm)**	73.0	9.1	72.0	12.5

**Table 3 sensors-22-07240-t003:** Statistics summary of EDQ metrics: ECG event signal ratio metrics (QRS SNR, PqrsR%, TqrsR%), results summary on the SAECG of the six bipolar arm-leads and on the reference Chest- Lead, over the 18 subject cases, after outliers removal.

Lead #	QRS SNR				PqrsR [%]				TqrsR [%]			
	Mean	SD	Median	IQR	Mean	SD	Median	IQR	Mean	SD	Median	IQR
**Chest**	996	735	741	1050	12.80	4.85	12.37	9.01	29.3	13.9	22.7	18.8
**Lead 1**	230	197	144	297	5.98	2.04	6.26	2.43	43.6	25.1	35.1	25.5
**Lead 2**	212	198	145	225	6.70	2.48	6.51	3.04	46.7	26.7	41.9	35.0
**Lead 3**	120	90	106	81	8.35	4.65	6.72	5.16	29.9	22.9	23.9	6.1
**Lead 4**	130	90	113	102	7.79	4.11	6.50	5.29	27.9	20.7	24.8	23.4
**Lead 5**	244	203	171	312	6.20	2.45	6.73	3.95	38.4	21.5	37.0	24.5
**Lead 6**	218	233	117	134	7.03	3.01	6.75	3.38	43.6	24.1	33.9	25.9

**Table 4 sensors-22-07240-t004:** Statistics summary of PT based arm-ECG QRS detection performance analysis results on the six bipolar arm lead derivations: mean value, SD, median and IQR results from the 18 subjects after outlier removal operation.

Lead #	Mean		SD		Median		IQR	
	Se%	PPV%	Se%	PPV%	Se%	PPV%	Se%	PPV%
**Lead 1**	86.8	99.3	15.7	0.8	93.3	99.6	15.7	0.6
**Lead 2**	75.6	98.8	27.9	1.7	85.1	99.6	32.3	1.1
**Lead 3**	41.5	96.3	36.5	4.6	54.1	98.6	65.7	6.3
**Lead 4**	53.9	97.3	37.1	3.9	72.3	98.7	73.0	2.2
**Lead 5**	86.1	98.5	15.5	1.9	92.2	99.3	11.9	2.4
**Lead 6**	68.8	97.5	29.1	2.9	79.8	98.2	27.5	3.2

**Table 5 sensors-22-07240-t005:** Results summary of Pearson Correlation coefficient (p) and scatter plot (Figure 7) associated trendline coefficient of determination (R^2^) between number of valid QRS events in the Chest Lead I (NVChestQRS) and the number of true positive QRS (NTPqrs) events on the six bipolar arm leads.

	Lead-1	Lead-2	Lead-3	Lead-4	Lead-5	Lead-6
**Pearson correlation (p) between: NVChestQRS vs. Arm leads NTPqrs**	0.88	0.86	0.30	0.44	0.83	0.66
**Scatter plot (Figure 6) trendline related Coefficient of Determination (R^2^)**	0.77	0.74	0.09	0.19	0.68	0.44

**Table 6 sensors-22-07240-t006:** Results summary of Pearson correlation coefficient (p) and scatter plot (Figure 9) associated trendline coefficient of determination (R2) between HRV RMS metric (of sinus beats IBI) measured on each of the six bipolar arm leads and on the Chest Lead I ECG (as the gold standard).

	Lead-1	Lead-2	Lead-3	Lead-4	Lead-5	Lead-6
**Pearson correlation (p) between: Chest RMS (HRV) vs. Arm-Lead RMS**	0.97	0.96	0.26	0.89	0.99	0.97
**Scatter plot ( Figure 9) trendline related Coefficient of Determination (R^2^)**	0.95	0.93	0.07	0.80	0.98	0.93

## Data Availability

Research data related to this article is available in the Ulster University PURE repository portal, under the name of the corresponding author’s research output, datasets area: https://pure.ulster.ac.uk/en/persons/omar-escalona/publications/.

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
