# Peer review of "Armband Sensors Location Assessment for Left Arm-ECG Bipolar Leads Waveform Components Discovery Tendencies around the MUAC Line"

_sensors, 2022, doi:10.3390/s22197240_

Round 1

Reviewer 1 Report

The concept behind this project is sound. Current use of continuous heart rate monitoring is based on PPG technology and I fully agree with the authors that continuous ECG monitoring is desirable, as long as it is both reliable and comfortable

Author Response

Sensors 1917464  (Arm-ECG manuscript, Round-1) Response to Reviewer 1 comments

___________________________________________________________________

Reviewer-1 Comments and Suggestions for Authors

1 -  The concept behind this project is sound. Current use of continuous heart rate monitoring is based on PPG technology, and I fully agree with the authors that continuous ECG monitoring is desirable, as long as it is both reliable and comfortable.

   Response from authors:

   The authors would like to thank this reviewer for their invested time and work on reviewing the manuscript, and for their appreciation of the underlying concept, highlighted clinical implications and for their positive feedback on the reported project.

Reviewer 2 Report

The paper presents "Evaluating the position of the armband sensor for the left Arm-ECG bipolar leads" Research to improve the quality of the obtained ECG signal which is often heavily affected by noise caused by muscle movement.

 Reviewers have some comments as follows:

1. The methods of testing and evaluation are presented in quite a detail and complete

2. Research results are presented clearly and fully analyzed.

3. The reviewer also has some suggestions:

3.1 The authors should revise the introduction to clarify the contributions in this paper.

3.2 The authors should add a small paragraph to present the layout of the article.

3.3. The quality of figures should be improved for it easier to see the number.

Author Response

Sensors 1917464  (Arm-ECG manuscript, Round-1) Response to Reviewer 2 comments

Review-2  Comments and Suggestions for Authors

1 - The paper presents "Evaluating the position of the armband sensor for the left Arm-ECG bipolar leads" Research to improve the quality of the obtained ECG signal which is often heavily affected by noise caused by muscle movement.

    Response from authors:

   The authors thank this reviewer for their devoted time and work on reviewing the manuscript and providing helpful feedback comments and suggestions for improving the manuscript.

 Reviewers have some comments as follows:

2 - The methods of testing and evaluation are presented in quite a detail and complete

    Response from authors:

   The authors are delighted to know the reviewer’s satisfaction and perspective opinion on the testing methods and evaluation aspects presented in the manuscript.

3 -  Research results are presented clearly and fully analyzed.

    Response from authors:

   The authors are delighted and grateful for this reviewer’s opinion about the Results presentation in the manuscript.

The reviewer also has some suggestions:

4 -  The authors should revise the introduction to clarify the contributions in this paper.

   Response from authors:

   The authors are grateful for this important observation about the Introduction part. This has been addressed by revising the last two paragraphs of the Introduction Section. But more particularly, the main contributions of this paper are clearly and briefly presented in the last paragraph, which was specifically edited to address this particular suggestion from the reviewer.

5 -  The authors should add a small paragraph to present the layout of the article.

   Response from authors:

   The authors have reworded the 2nd to last paragraph of the Introduction Section to address this suggestion from the reviewer and avoid possible repetition of content; if a separate paragraph was added. Along that penultimate paragraph of the Introduction, the layout of the article has been emphasised by indicating the Section numbers of main content parts in the manuscript; within the Methods & Materials, Results and Discussion sections of the article.

6 -  The quality of figures should be improved for it easier to see the number.

   Response from authors:

   Thank you for this valuable observation to improve the easy viewing/reading directly and without having to zoom in and out, etc. Therefore, the font size of some legend text and of both axes (x and y) scale numbers in Figure 4, and in figures 6 to 9, have been increased considerably to improve the quality of figures to enable easy reading of text and numbers in the figures.

Reviewer 3 Report

The manuscript presents a comprehensive analysis aiming to identify ECG bipolar electrode pairs' best vectorial orientation around the left upper arm. The novelty of the proposed research is supported by the lack of literature in this field of study. I really appreciate the ECG signal processing part and the analyzed sample size. The presented study is of great potential interest to readers interested in implementing a wearable ECG solution in a more unobtrusive way than conventional ECG Holter. From the writing point of view, the manuscript must be checked for typos, and the grammatical issues should be improved. Nevertheless, I do not have any remarks about the formal part of the manuscript, and I strongly recommend publishing it.

Author Response

Sensors 1917464  (Arm-ECG manuscript, Round-1) response to Reviewer 3 comments

Reviewer-3 Comments and Suggestions for Authors

1 -  The manuscript presents a comprehensive analysis aiming to identify ECG bipolar electrode pairs' best vectorial orientation around the left upper arm. The novelty of the proposed research is supported by the lack of literature in this field of study. I really appreciate the ECG signal processing part and the analyzed sample size. The presented study is of great potential interest to readers interested in implementing a wearable ECG solution in a more unobtrusive way than conventional ECG Holter.

   Response from authors:

   The authors are most grateful to this reviewer for the time and effort invested in reviewing the work presented in this manuscript. Also, the authors much appreciate their clear interpretation of the processing methods and analysis in the presented study, as well as highlighting the value and importance of this work in knowledge contribution for advancing a new approach to wearable ECG monitoring.

2 -  From the writing point of view, the manuscript must be checked for typos, and the grammatical issues should be improved.

   Response from authors:

   The authors have conducted a thorough check for typos and grammatical issues, and indeed, have spotted several of these and corrected them, as revealed in the MS Word revised manuscript with “Track Changes” enabled. A major detected typo was on the figure numbering of the last figure, which now is Figure 10, and all referencing instances to that figure in the main text body were amended accordingly.

   Nonetheless, the authors would like to indicate that the manuscript was written in UK English language (spelling and grammatical), hence some possible words with different spelling as in US English, e.g., centre (in UK English) and center (in US English), etc.

3 -  Nevertheless, I do not have any remarks about the formal part of the manuscript, and I strongly recommend publishing it.

   Response from authors:

   Many thanks for your supporting positive comments about the manuscript and its publication recommendation.
